# Comprehensive Assessment of Environmental Pollution in a Poultry Farm Depending on the Season and the Laying Hen Breeding System

**DOI:** 10.3390/ani12060740

**Published:** 2022-03-16

**Authors:** Tomasz Szablewski, Kinga Stuper-Szablewska, Renata Cegielska-Radziejewska, Łukasz Tomczyk, Lidia Szwajkowska-Michałek, Sebastian Nowaczewski

**Affiliations:** 1Department of Food Quality and Safety Management, Poznan University of Life Sciences, 60-624 Poznań, Poland; tomasz.szablewski@up.poznan.pl (T.S.); renata.cegielska-radziejewska@up.poznan.pl (R.C.-R.); lukasz.tomczyk@up.poznan.pl (Ł.T.); 2Department of Chemistry, Poznan University of Life Sciences, 60-624 Poznań, Poland; kinga.stuper@up.poznan.pl; 3Department of Animal Breeding and Product Quality Assessment, Poznan University of Life Sciences, Słoneczna 1, 62-002 Suchy Las, Poland; sebastian.nowaczewski@up.poznan.pl

**Keywords:** bioaerosols, laying hens, litter, volatile compounds, pollutants

## Abstract

**Simple Summary:**

The source of odors and dust emitted from hen houses are elements of the poultry house environment, such as litter, feed, and animals. The concentration of volatile compounds and the composition of the microflora depend on the hen farming system and the season. The research carried out as part of this study is a comprehensive assessment of the microbiological contamination (*Pseudomonas*, *Enterobacteriaceae*, and microscopic fungi) of all the elements that make up the environment of the poultry house in an annual cycle. Two types of laying hens reared on litter were compared: commercial and backyard farms. It was found that the seasons of the year and the system of keeping hens have a significant impact on the microbiological contamination with volatile compounds of the environment and the atmosphere of the hen houses. The obtained results of chemical, microbiological and questionnaire tests show that commercial farms carry a lower microbiological risk to the environment than backyard farm.

**Abstract:**

The odors and dust emitted from hen houses affect human health and the condition of crops. The source of fumes is an element of the poultry house environment that affects the level of dust (litter and feed), the concentration of volatile compounds and the composition of the microflora (litter, dust and fodder). The research carried out as part of this study is a comprehensive assessment of the microbiological contamination (*Pseudomonas*, *Enterobacteriaceae*, and microscopic fungi) of all the elements that make up the environment of the poultry house (feed, litter, dust pollution and the atmosphere of the poultry house) in an annual cycle. The air from both types of farms is tested in terms of the quantity and quality of volatile compounds. Two types of laying hens reared on litter were compared: commercial and backyard farms. It was found that the seasons of the year and the system of keeping hens have a significant impact on the microbiological contamination with volatile compounds of the environment and the atmosphere of the hen houses. The obtained results of chemical, microbiological and questionnaire tests show that commercial farms carry a lower microbiological risk to the environment than backyard farm.

## 1. Introduction

Activities for the protection of the environment, which focus on minimizing the negative effects of human activities on the environment, have recently become an important element of the agri-food industry. The Act of 27 January 2001 on the Environmental Protection Law (Journal of Laws of 2001, No. 62, item 627) [1] contains provisions on environmental protection and sustainable development. When planning to launch intensive livestock production, including commercial farms, various methods are used to reduce the emission of harmful compounds to the environment.

According to the data of the Central Statistical Office (2020), in 2019, over a half of farms (57.8%) were those with an average area of less than 5 ha. Therefore, it is characteristic for Poland, due to the presence of many small backyard farms, to keep hens in the homestead system to meet their own needs. The species structure, according to statistical data for December 2011, consists of chicken poultry in 92%, of which 32.8% are laying hens. In order to increase productivity, profitability and profits, poultry is kept more and more often on commercial farms. A total of 30% of bedding is used on laying hen farms in Poland and on broiler farms [2].

In the literature about poultry farming, there are reports on research conducted on intensive poultry farms, where the impact of emitted fumes from poultry houses on the surrounding environment was analyzed [3,4]. However, this risk has not yet been investigated for backyard farms where fewer layers are kept, and control of the farming conditions is limited. The pollutants of the henhouse atmosphere are discharged outside by two methods: forced (mechanical ventilation system) and gravity (forced air circulation), both used in farms. The pollutants emitted in this way mainly consist of organic and inorganic dust that create bioaerosols that are hazardous to human and animal health [5,6,7]. The presence of microorganisms and chemical compounds in the atmosphere of the poultry house is important. As shown by the research of Hartung and Schulz [4], pollutants emitted from animal farms pose a threat to the respiratory system of humans and animals, the environment, including soil and water, and cause global warming. Bioaerosols contained in the air also affect the health of farmers and residents of the neighboring area. Intensive livestock production generates very large amounts of harmful chemical compounds into the environment in the form of harmful gases, wastewater, dust and pathogens, including bacteria and fungi [5,8,9,10].

The aim of the study is to assess the environmental risk associated with the breeding of laying hens kept in commercial and backyard farms.

This goal is achieved through the implementation of the following research tasks:The determination of changes in the level of microbiological contamination of the elements that make up the environment of poultry houses depending on the season;The determination of changes in the level of air pollution of poultry houses with volatile compounds (odorants) depending on the season;The assessment of the risk to the external environment related to the emission of dust fumes from poultry houses;The assessment of the nuisance of the neighborhood of hen houses in relation to the inhabitants.

## 2. Materials and Methods

### 2.1. Test Material

The tested material consisted of air, dust, litter and fodder samples taken from 2 systems of keeping laying hens on the litter (16 hen houses in backyard farms and 8 hen houses in commercial farms). The samples were taken annually, i.e., in spring, summer, autumn and winter, in three repetitions from each hen house. The first system is a traditional, backyard farms with a small number of birds (less than 1 head/m^2^), with a private enclosure available in spring, summer and autumn. The hens were 60 weeks old at the beginning of the sampling. These were Rhode Island Red hens kept on straw bedding. Hen houses were situated in a village inhabited by over a thousand inhabitants in Poland (Table 1). The livestock buildings were located in the very center of the town, directly adjacent to other buildings, including residential buildings. In the case of the second system of keeping hens, the hens came from 8 commercial breeding hen farms. Each of the farms consisted of poultry houses where the Cobb and Hubbard Flex parent flocks were reared. The size of the hen houses was larger and amounted to 1200 m^2^ for the first livestock building, and 1000 m^2^ for the second. The stocking of poultry houses was also greater, as per 1 m^2^, it was on average 6 pcs/m^2^. In addition, the livestock buildings were fully mechanized, with automatic feeding belts and water drinkers. In addition, to provide the birds with appropriate living conditions, mechanical ventilation is used, consisting of 6 fans. The poultry was kept on the bedding, which consisted of straw entirely in the rearing house and in a short-cut form at the later stages of rearing.

### 2.2. Sampling Methods

The samples were taken four times in an annual cycle, i.e., in winter, winter, summer and autumn. Samples weighing 20 g were taken.

#### 2.2.1. Feed and Bedding Intake

Sampling for solid elements, such as feed and bedding, consisted of collecting representative samples by means of a manual method using the isolation method.

#### 2.2.2. Dust Collection

Dust samples were collected using the gravimetric method in accordance with PN-EN 13284-1: 2007 “Emission from stationary sources. Determination of the mass concentration of total dust, Gravimetric method” [11].

#### 2.2.3. Volatile Metabolites

Volatile metabolites were separated from the hens’ atmosphere by microextraction into the solid phase, which in this case was a 100 μm thick layer of polydimethylsiloxane (PDMS) bound on a carrier fiber. Fiber was placed in the center of the house 1 m above the floor and exposed for 30 min. The sample was taken in 3 replicates.

### 2.3. Laboratorial Analysis

The research carried out as part of this study was a comprehensive assessment of the microbiological contamination of all the elements that make up the environment of the poultry house (feed, litter, dust pollution and the atmosphere of the poultry house) in an annual cycle. The air from both type of farms was tested in terms of the quantity and quality of volatile compounds. Two types of laying hens reared on litter were compared: commercial and backyard farms.

#### 2.3.1. Total Number of Bacteria

To determine the total bacteria count (TBC), the following procedure was followed. They were fragmented and standardized. The weight of a single sample for analysis was 10 g. Initially, 10 g of the ground test material from each test was suspended in 90 mL of the diluting fluid (Merck KGaA, Darmstadt, Germany). Then, tenfold dilutions were made from the prepared suspensions in the diluting fluid. Inoculations were performed up to 20 min from the preparation of the solutions. To this end, 1 mL of the suspension from the two dilutions of each sample was first transferred with a sterile pipette to sterile Petri dishes (two for each dilution), and then they were flooded with 15 mL of agar medium (BTL nutrient agar, Merck KGaA, Darmstadt, Germany) at 45 °C. The prepared plates were incubated in aerobic conditions, placed flat in an incubator at the temperature of 30 ± 1 °C for 72 h. After incubation, the bacterial colonies on all plates were counted, and based on the number of colonies counted, the total number of bacteria in 1 g of test material was obtained (CFU/g). The final result was the mean and was expressed as log CFU/g.

#### 2.3.2. Enterobacteriaceae Count

The assessment of the number of *Enterobacteriaceae* bacteria per 1 g of feed was performed in accordance with the test procedure, the standard decimal dilution plate method (PN ISO 21528-2: 2005 Microbiology of food and fodder).

#### 2.3.3. Pseudomonas Count

To detect the number of bacteria of the genus *Pseudomonas* [12], the method was identical to that for the determination of the total number of bacteria (Section 2.3.1) with a difference in incubation time, which for the determination of the number of *Pseudomonas* bacteria was 48 h.

#### 2.3.4. The Number of Molds and Yeasts

The assessment of the number of molds and yeasts per 1g of feed was performed according to the test procedure using the standard decimal dilution plate method (PN-ISO 21527-2: 2009. Microbiology of food and feed. Horizontal method for the determination of the number of yeasts and molds. Part 2: Colony counting method in products with a water activity lower than or equal to 0.95) [13]. The diluted method was used: 1 g of sample was put in 10 mL of sterile distilled water and mixed with the magnetic stirrer for 2 min. Next, 1 mL of suspension was carried on potato-dextrose agar medium (BTL, Lodz, Poland) in Petri dishes and spread on the medium surface with a sterile glass stick. The Petri dishes were incubated at 25 °C for 7 days.

#### 2.3.5. Volatile Compound Analysis

Volatile compound analysis was performed using a gas chromatograph (Hewlett Packard 6890) equipped with a mass detector (Hewlett Packard 5972 A) and an HP-5MS column (30.0 m, 0.25 µm). In order to identify the volatile compounds, the analysis was carried out in the mass range of the spectrometer 50–250 *m*/*z*. The mass spectra of the peaks obtained during the run were compared with the mass spectra contained in the NIST02 and Wiley 7N library or original standards. On the basis of the identified compounds, they were assigned to 10 groups: alcohols, aldehydes, ketones, hydrocarbons, acids, terpenes, benzene and its derivatives, phenols, and sulfur-containing compounds. The remaining volatile compounds that did not belong to any of the above-mentioned groups were combined into the “other” group.

### 2.4. Survey Method

In the first stage of the study, a detailed questionnaire was developed, which, after making the necessary corrections in the preliminary study, was duplicated in an appropriate number of copies.

The obtained results were analyzed with the following non-parametric tests:Tests of differences between groups (independent samples);Tests of differences between variables (dependent samples);Tests of correlations between variables (Spearman’s correlation, chi-squared).

The questionnaires included single-choice and multiple-choice closed questions as well as open-ended questions. The conducted surveys were aimed at acquiring knowledge about the environmental nuisance for employees and for the residents neighboring the henhouses. The obtained answers were summarized using basic mathematical and statistical methods and elaborated on the charts.

### 2.5. Statistical Analysis

The results obtained in the course of the conducted chemical analyzes were statistically analyzed in the STATISTICA v 8.0 software. In order to compare the content of individual metabolites and the level of contamination with microorganisms, the Tukey’s method of multiple comparisons was used at the significance level of α = 0.05. The analysis of the survey results was made on the basis of a non-parametric comparison of independent samples with the Mann–Whitney U test.

## 3. Results

As part of this study, samples of air, dust, litter and feed collected from 2 systems of keeping laying hens on litter were analyzed (16 hen houses from backyard farms and 8 from commercial farm hen houses).

### 3.1. Microbiological Contamination

In the framework of this study, the settled dust collected from the inside of the house and from the outside was analyzed. The microbiological contamination of the settled dust samples was assessed. On the basis of the obtained results, it was found that, among the analyzed groups of microorganisms, there were significantly more bacteria than microscopic fungi (Table 2). The smallest group among the analyzed microbiological contaminants was that of the *Enterobacteriaceae* family, which did not occur in the farm breeding system during winter, and in subsequent seasons, it increased to a maximum of 4.5 × 10^5^ in autumn. In the case of large-scale farming, these values were lower than in the farm-side system and ranged from 5.9 × 10^2^ in spring to 7.2 × 10^3^ in autumn. Contamination with microorganisms from the *Pseudomonas* family was similar in both farming systems. The amount of mold and yeast contained in the settled dust varied depending on the season and the rearing system. The lowest contamination with microscopic fungi in large-scale farming occurred in autumn (5.0 × 10^4^) and the highest in winter (1.4 × 10^8^). In the system of farm breeding, these values ranged from 3.3 × 10^6^ in winter to 5.0 × 10^7^ in autumn.

Table 3 shows the mean microbial contamination for dust collected outside the house. The total number of bacteria contained in the dust collected outside by the ventilator did not differ in the order of magnitude over the following seasons and was on average 3.5 × 10^8^ CFU/g. The level of contamination with microorganisms of the *Enterobacteriaceae* family gradually increased, starting from 1.1 × 10^2^ in winter and ending with 6.5 × 10^3^ in autumn. A similar upward trend can be noticed in the case of microorganisms from the *Pseudomonas* family, where in winter the CFU/g number was 1.4 × 10^6^ to 1.6 × 10^8^ in autumn. The number of molds and yeasts determined in the large-scale farming system was different. The lowest contamination with microscopic fungi was identified in winter (1.1 × 10^4^) and the highest in autumn (6.1 × 10^7^).

The mean microbiological contamination of the litter collected from both houses varied (Table 4). The highest number of CFU/g was determined in the case of TBC in the summer large-scale farming system (3.3 × 10^11^). The order of magnitude of the total number of determined bacteria was similar in both farms and amounted to an average of 4.0 × 10^10^ for farmyard breeding and 1.9 × 10^11^ for large-farm breeding. Litter contamination with *Enterobacteriaceae* bacteria was higher in the farmstead system (3.3 × 10^4^ to 1.8 × 10^6^) than in the large-farm system (3.6 × 10^3^ to 5.3 × 10^5^). On the other hand, the amounts (CFU/g) of the *Pseudomonas* family were reversed, where the lowest content of these microorganisms was determined in summer and spring in the farmstead system (i.e., 1.8 × 10^6^ and 4.8 × 10^6^, respectively) and the highest in winter (1.8 × 10^10^). The lowest values of contamination with microflora from the *Pseudomonas* family in the case of large-scale farming were observed in autumn (3.8 × 10^7^), and the highest, similarly to the farmstead system, in winter (3.2 × 10^10^). The highest concentration of mold was found in autumn in farmyard rearing (2.6 × 10^8^), and the lowest in summer in the commercial farming system (1.5 × 10^4^).

The last analyzed element of the poultry house environment was the feed. A summary of the average content of individual contaminants contained in the feed is presented in Table 5. The highest contamination of the feed among all the analyzed groups was determined in the farm breeding system in spring (5.4 × 10^11^) for TBC, and in the large-farm system in winter (4.5 × 10^10^). The smallest share in microbiological contamination had the microflora of the *Enterobacteriaceae* family, which in the large-scale farming system was 5.3 × 10^1^ in summer and 7.3 × 10^1^ in autumn. The same was the case in the homestead system, where the number of CFU/g in spring was 2.8 × 10^2^ and in autumn 2.3 × 10^3^. The group of microorganisms from the *Pseudomonas* family also had a significant share in the overall contamination of the feed, which was the maximum of 1.6 × 10^10^ in the spring in the farmyard system and 8.8 × 10^9^ in the summer in the large-farm system. The content of mold and yeast in the fodder used in the farmstead system ranged from 1.4 × 10^4^ in summer to 2.7 × 10^6^ in autumn. In the case of feed used in large-scale farming, the values ranged from 3.3 × 10^3^ in autumn to 7.4 × 10^6^ in spring.

### 3.2. Volatile Compound Content in the Hen House Atmosphere

As part of this study, 94 volatile compounds were identified for the farm system and 66 for the commercial farming system. Table 6 and Table 7 present the results of the obtained volatile compounds in the system of farm and large-scale farming, taking into account the seasons of the year. The concentration of individual chemical compounds varies significantly depending on the system of hen rearing and depending on the season. The greatest variety of volatile compounds was determined in summer and autumn. In winter, the air in the chicken coop was the least polluted, so some compounds were not detected at this time of the year. In terms of quality, the most numerous group was that of alcohols, 26 of which were identified. The smallest groups containing four different volatile compounds were terpenes, phenols and sulfur-containing compounds. The highest concentrations of average contents for individual groups of compounds were found in acids and ketones, both in the farm and commercial farming systems. The lowest concentration of average contents was recorded for terpenes, benzene and its derivatives, and hydrocarbons. The values of individual compounds changed depending on the season. In the majority of cases, the means for samples taken in the system of commercial or barn farming differed significantly depending on the season of the year (*α* = 0.05).

By only comparing the average content of individual groups of volatile compounds in the two analyzed systems of laying hen rearing (i.e., farm and commercial farming), we can observe significant differences (α = 0.05) for groups, such as alcohols, aldehydes, ketones, acids, sulfur-containing compounds and other. The other groups do not differ significantly with regard to the rearing system (Table 6). On the basis of the obtained quantitative results of volatile compounds in the air, it can be concluded that commercial farming is worse. However, the qualitative picture of volatile compounds indicates a greater diversity in their occurrence in backyard farming, which carries a greater risk.

For all groups, the mean concentrations for sampling did not differ significantly in the farm and commercial farming systems, with the exception of benzene and its derivatives in summer and autumn, and for sulfur-containing compounds from spring and summer trials (Table 7).

When analyzing the concentration of particular groups of compounds for the farm and large-scale systems, an upward trend can be observed for individual seasons, starting from winter to autumn. On this basis, the formation curves were determined. The concentration for the average content of alcohols, ketones, acids, terpenes, phenols, sulfur-containing compounds and remaining in the large-scale farming system over the changing seasons of the year was higher than for farm breeding. The opposite was the case with aldehydes, hydrocarbons and benzene derivatives, where higher values were recorded in farm farming than in large-scale farming. On the basis of the obtained quantitative results of volatile compounds in the air, it can be concluded that commercial farming is worse. However, the qualitative picture of volatile compounds indicates a greater diversity in their occurrence in backyard farming, which carries a greater risk.

Among all determined volatile compounds, those produced by the metabolism of microscopic fungi are of significant importance [14,15]. The content of trichodiene gradually increased with the change of the season (Figure 1). Summing up, its higher concentration was found in the environment of poultry houses in which hens were reared in the farm, compared to large-scale farming (Figure 2).

### 3.3. Survey Results

#### 3.3.1. Surveys Conducted among Employees of Commercial Poultry Houses

The respondents are mainly employees who have been practicing this profession for over 5 years. The observed pollutants on the farm are assessed mainly as an average, which occurs periodically. Mostly, odorous impurities, i.e., odors, and solid impurities, such as dust and dirt, were distinguished. The surveyed employees provided information about the lack of influence of these pollutants on the condition of the animals. In addition, no mycosis of the respiratory tract, beak thrush or ulceration was observed in chickens. In the case of the majority of responses, the respondents did not observe dementia in chickens and birds that were swollen above and below the eye. However, sneezing was noticed in them, which occurred sporadically and intensified with high humidity and dustiness in the air. Farm workers did not observe eye rubbing against feathers, difficulty breathing or nasal discharge in the hens. The most frequently given answers to the assessment of the fumes emitted on a 5-point scale are 3 and 4 points. Most of the respondents stated that the intensity of odors emitted from hen houses is variable. The intensity of these odors is felt most in summer, then in autumn and spring. Winter turned out to be the least troublesome time of the year.

Employees who were interviewed often deal with allergies, asthma and skin diseases. A large proportion of respondents admit that they experience burning and itching of the eyes as well as coughing, which intensify in spring, summer and autumn. These symptoms worsen when staying in hen houses and outdoors. Most of the respondents were non-smokers. Moreover, they are accompanied by headaches, which also occur without any obvious symptoms, such as fever and runny nose, that is, without any apparent cause. The headaches of which the respondents complained intensify during the stay in the livestock buildings and after leaving them. Workers also experience recurring migraines. The main chronic diseases mentioned in the respondents are allergy and asthma.

#### 3.3.2. Surveys Conducted among Residents Neighboring with Commercial Henhouses

Additional material for the assessment of environmental nuisance were surveys conducted among neighbors and employees of the poultry farm. A total of 340 residents adjacent to large-scale hen houses were interviewed. A total of 150 people completed the surveys among employees of poultry farms. The age of the respondents was in the 31–70 age range. A total of 63% of the respondents were women and 27% were men. All respondents were under constant health care and had checkups performed more than once in 5 years. The surveyed neighbors mainly (62.5%) live in single-family houses with a garden, 25% in single-family houses with an average area of 2.5 ha., and 20% of the respondents lived in a single-family house without a garden or farm. In the case of 87% of respondents, when they started to live in their current place of residence, there were already hen houses in their vicinity. Only 13% of the respondents lived in this place before the new hen houses. More than half of the respondents (57%) did not realize that being in the vicinity of hen houses may have an impact on the comfort of their living, but 43% were aware of it. Despite the uncomfortable living conditions resulting from being in the vicinity of hen houses, nearly 90% (88%) of the respondents do not plan to change their place of residence. Over 10% of respondents considered the possibility of changing their place of residence. A total of 50% of the respondents assessed the condition of their local environment as good. On the other hand, 25% of residents described it as good, as did another 25% who considered the condition of the environment as bad. The respondents assessed the degree of air pollution in their city similarly to the condition of the environment in question 7. For 50% of the respondents, the air pollution was considered to be medium and 25% of the respondents described the air condition as good or bad. However, none of the respondents chose the answer that would indicate a complete absence of air pollution in the place of residence. When asked about the frequency of air pollutants in their localities, 50% stated that they appear periodically. According to 13% of respondents, air pollution appears sporadically, while 37% describe the frequency of their occurrence as permanent. Half of the respondents indicated odors emitted from neighboring hen houses as the main type of pollution. Over 36% considered air pollution with solid particles, such as dust and dirt. More than 20% mentioned other types of air pollution, such as soot, odors from sugar factories and fat production, burnt garbage or faulty sewage systems. Selected response statistics are shown in Figure 3.

The survey research carried out showed that being in the vicinity of a large-scale hen house is burdensome for the inhabitants and the environment due to the presence of odorants in the air and high dustiness of the air, contributing to plant diseases, mainly fungal diseases, and the presence of allergies among the inhabitants and employees of the henhouse.

## 4. Discussion

On the basis of the conducted research, several significant relationships can be observed between the season of the year and the content of chemical and microbiological pollutants, which contradict the general opinion of the society.

The number of identified volatile compounds in the large-scale farming system is lower than in the backyard-farm farming system. Some of the identified compounds, despite their small quantitative share, have a significant impact on the perceived nuisance associated with a strong odor-generating effect. Long-term exposure to harmful volatile compounds and bioaerosols may have a negative impact on the health of employees [8,16,17] as well as neighboring neighbors. Moreover, the increased inconvenience to health is related to the composition of bioaerosols in the air in which these people live [18]. As it results from the surveys carried out in this study, both in the case of people working on the poultry farm and those close to the farm, there are frequent headaches, runny nose, burning eyes, itching of the skin and other symptoms that are not directly related to specific disease conditions, but they occur chronically. This may result from the exposure of the human body to the harmful effects of many different compounds [19]. The concentrations of individual groups of volatile compounds increase with the change of the season. The lowest level of air pollution with volatile compounds occurred in winter, and then it increased, reaching the maximum concentration in summer and autumn. The atmospheric conditions during sampling, such as temperature, humidity, as well as the force and direction of the wind, may have influenced the formation of volatile compounds in the air [5]. High air humidity, which occurs especially in summer and autumn, and higher temperature favor the development of microflora, which in such conditions multiplies rapidly. Its presence poses a direct threat to health, as do metabolites, which are produced by microorganisms [3,20,21]. The increase in the perception of the emitted odors may be related, in particular, to the increase in temperature [22].

Many publications emphasize the influence of microorganisms and chemical compounds produced during animal production, which must be observed in order to ensure the welfare of animals and create safe conditions for people staying in highly dusty rooms that contain a mixture of various substances [3,5,7,21].

The dust inside the large-scale hen house contained a larger number of microorganisms than in the pen-and-farm system, which resulted from the high density per 1 m^2^ and the total number of birds in the hen house, as well as high humidity and temperature, creating a microclimate favorable for microflora to multiply [23]. However, the amount of microbiological contamination contained in the dust collected inside the hen house in farmyard farming did not differ significantly from the results obtained in large-scale farming.

The total number of bacteria in summer (9.2 × 10^10^) was the largest in the large-scale farming sector, similarly to the TBC system of farm breeding in summer, which was the highest (1.5 × 10^8^).

The content of microflora in the dust settled outside the poultry house in the large-scale system largely coincided (approx. 80%) with the values of individual groups of microorganisms identified in the dust inside the poultry house. These values differed due to differences in the weather conditions outside the house, which made the values lower. This phenomenon is related to the action of sunlight, low or high temperatures, rainfall or lightning, which help to purify the atmosphere of microbial contamination [24,25].

Microbiological contamination of litter in both analyzed systems of laying hens rearing was diversified and could result from the period of manure remaining in the henhouse. Nevertheless, the highest concentration of microbiological contamination of the litter, as in the case of dust settled inside, occurred in spring and summer, which may be related to the prevailing temperature and humidity conditions. Determining unambiguous relationships between the microbial contamination of litter and the season of the year is difficult due to the lack of information on the dates of removing contaminated litter from poultry houses. They obtained similar results (Stuper-Szablewska et al. [26]).

In the case of microbiological contamination determined in the feed used for poultry feeding, the obtained values are variable. The differences between the seasons of the year in a multi-farm system may be due to different batches of feed that may have already been initially microbiologically contaminated. The feed delivered to the commercial farm was monitored and standardized on an ongoing basis, making it easy to check where it came from, and where and when it was purchased. In the case of farm breeding, the fodder most often comes from its own cultivation and is not subject to testing. Moreover, the highest concentration of microbial contamination in barns was observed in spring, which could be related to improper storage. In the autumn, the bacterial contamination decreased, while the contamination with microscopic fungi increased—this may indicate that the new batch of grain used for fodder was less bacterially contaminated, but contained more mold.

## 5. Conclusions

On the basis of the conducted research, it was found that large-scale farming carries a lower microbiological risk to the environment than backyard farm farming. It was found that the seasons of the year and the hen housing system have a significant impact on the microbiological contamination of the environment in the poultry house. The highest microbial contamination was observed in the village (bacteria) and in the fall (microscopic fungi and yeasts). In addition, the seasons of the year and the system of keeping hens also have a significant impact on the air pollution of poultry houses with volatile compounds (odorants). Based on the results of the microbiology of dust collected from the inside and outside of the poultry house, it was found that the removed dust has a negative impact on the environment adjacent to the poultry house. However, in the case of in-house henhouses where the dust is not removed from the inside of the house to a significant extent, the residual dust inside the house can cause bird diseases.

The survey studies carried out showed that the vicinity of a large-scale hen house is burdensome for inhabitants and the environment due to the presence of odorants in the air and high dustiness of the air contributing to mainly fungal diseases in plants and the presence of allergies in the inhabitants and employees of the poultry house.

The systematic control of the quality of air, feed, bedding and other components in poultry production can help to minimize the emitted fumes.

Summing up the obtained results, attention should be paid to the possibility of controlling the prevailing conditions in poultry houses, which is currently monitored very rarely in farmstead systems.

In order to minimize the risks arising during animal husbandry, both for industrial and individual purposes, it is important to systematically control the prevailing conditions. The microbiological and chemical monitoring of air pollution, feed, litter and other components can help to minimize the negative effects of their occurrence. Knowing the level of pollution, we can make decisions that will reduce the fumes emitted from poultry houses, and thus obtain better conditions for animal breeding, plant production, people working on farms and their neighbors.

## Figures and Tables

**Figure 1 animals-12-00740-f001:**
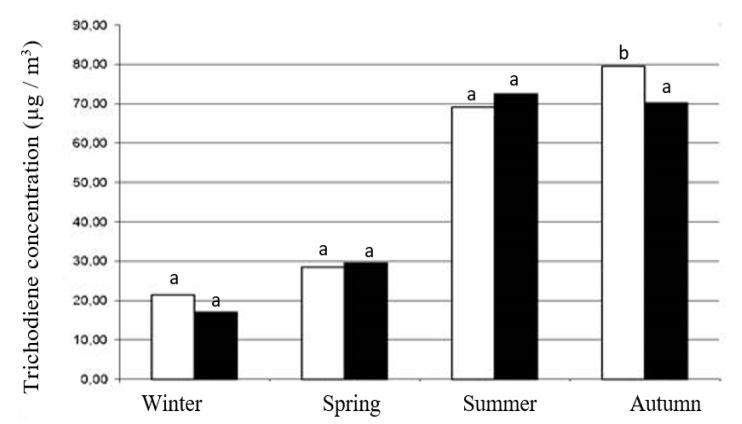
Trichodiene concentration (µg/m^3^) in the atmosphere of poultry houses (commercial farms: black; backyard farms: white) in the annual cycle. a, b—different in the row for each of the two analyzed chow systems; significance was set at the value of 0.05.

**Figure 2 animals-12-00740-f002:**
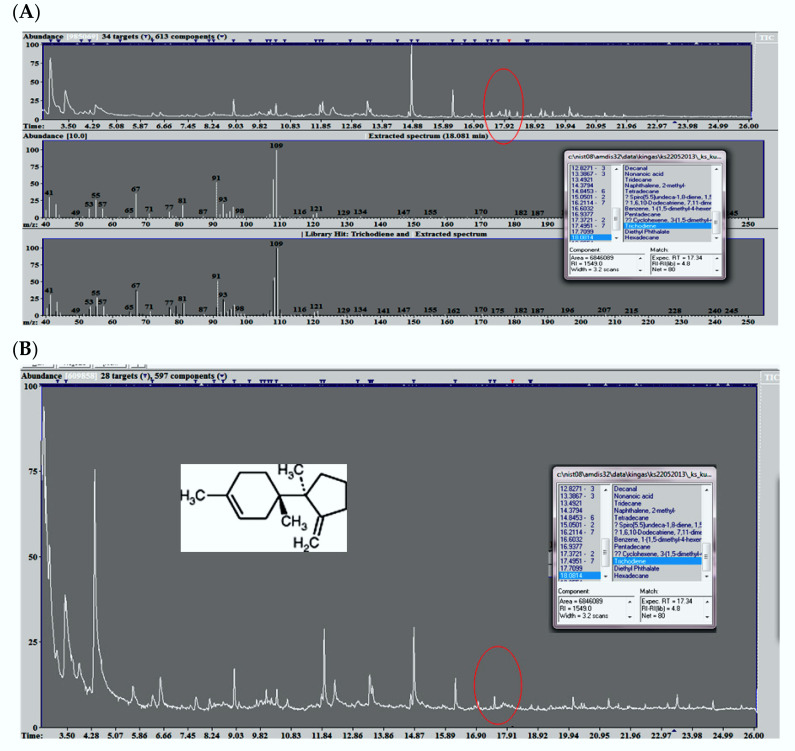
Examples of VOC chromatograms with particular emphasis on trichodiene (the circle marks the area on the chromatogram that allows for the interpretation of the analyzes) (**A**) commercial farming (chromatogram, mass spectrum) and (**B**) backyard farm breeding (chromatogram, structural formula).

**Figure 3 animals-12-00740-f003:**
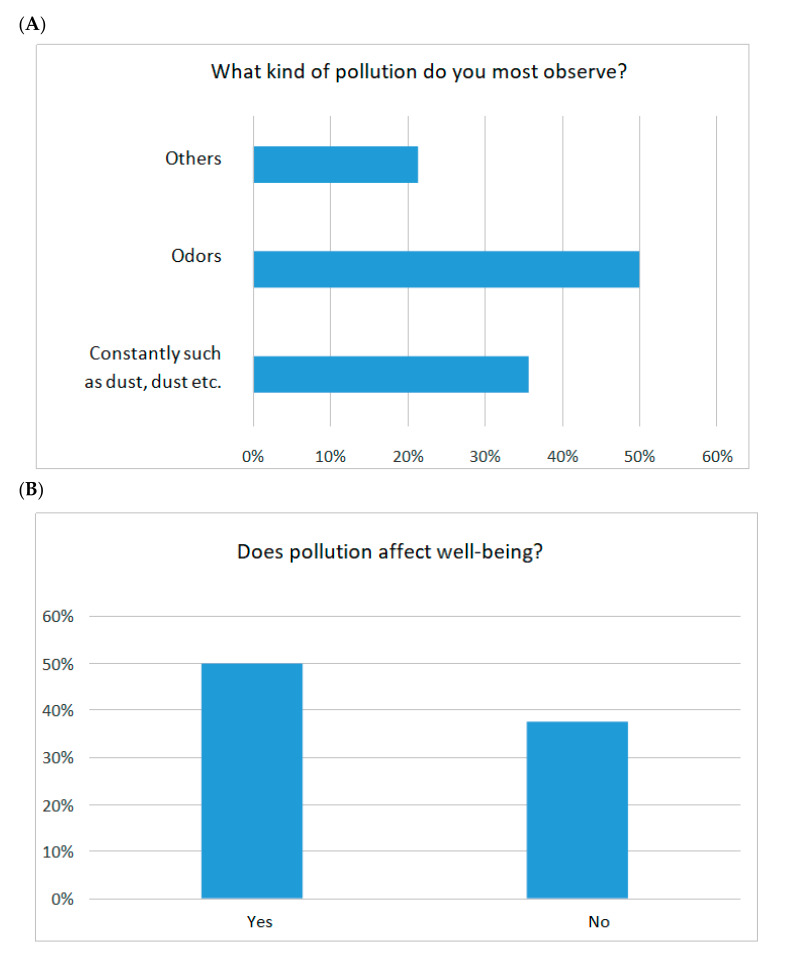
Selected statistics of the respondents’ answers to survey questions from residents neighboring with commercial hen houses. (**A**) What kind of pollution do you most observe? (**B**) Does pollution affect well-being? (**C**) What are the odors most noticeable.

**Table 1 animals-12-00740-t001:** Poultry house locations.

Lp.	Location	Province	Type of Poultry Production
1	53°31′56″ N 20°67′35″ E	Greater Poland Province	Commercial
2	52°01′93″ N 17°78′44″ E	Greater Poland Province	Commercial
3	52°65′67″ N 16°95′29″ E	Greater Poland Province	Commercial
4	51°62′26″ N 17°94′28″ E	Greater Poland Province	Commercial
5	52°99′64″ N 18°70′72″ E	Kuyavia-Pomerania Province	Backyard
6	49°39′96″ N 22°44′98″ E	Podkarpacie Province	Backyard
7	49°27′54″ N 19°86′88″ E	Lesser Poland Province	Backyard
8	51°25′05″ N 22°57′01″ E	Lublin Province	Commercial
9	51°29′03″ N 20°51′06″ E	Lublin Province	Commercial
10	50°29′68″ N 16°65′20″ E	Lower Silesia Province	Backyard
11	53°92′82″ N 14°44′89″ E	West Pomerania Province	Backyard
12	53°91′31″ N 14°52′00″ E	West Pomerania Province	Backyard
13	53°47′30″ N 17°89′64″ E	Kuyavia-Pomerania Province	Commercial
14	53°48′46″ N 18°07′17″ E	Kuyavia-Pomerania Province	Commercial
15	53°77′66″ N 20°47′65″ E	Warmia-Masuria Province	Backyard
16	53°39′84″ N 20°94′62″ E	Warmia-Masuria Province	Backyard
17	53°58′34″ N 20°28′16″ E	Warmia-Masuria Province	Backyard
18	51°24′99″ N 21°58′16′′ E	Lublin Province	Backyard
19	53°81′29″ N 20°35′80″ E	Warmia-Masuria Province	Backyard
20	54°86′21″ N 21°37′89″ E	Warmia-Masuria Province	Backyard
21	54°47′25″ N 16°63′07″ E	West Pomerania Province	Backyard
22	55°48′20″ N 16°77′01″ E	West Pomerania Province	Backyard
23	54°16′88″ N 17°49′22″ E	Pomerania Province	Backyard
24	54°22′32″ N 17°97′49″ E	Pomerania Province	Backyard

**Table 2 animals-12-00740-t002:** Average microbial contamination (CFU/g) of settled dust collected from inside the henhouse.

Type of Poultry Production	Season	TBC	*Enterobacteriaceae*	*Pseudomonas*	The Number of Molds and Yeasts
Backyard	Winter	1.6 × 10^6 a^	-	1.5 × 10^8 a^	3.3 × 10^6 a^
	Spring	2.5 × 10^5 b^	1.2 × 10^2 a^	6.7 × 10^7 b^	4.9 × 10^7 b^
	Summer	1.5 × 10^8 a^	1.5 × 10^4 a^	5.3 × 10^8 b^	4.4 × 10^6 a^
	Autumn	5.2 × 10^7 c^	4.5 × 10^5 b^	1.4 × 10^8 a^	5.0 × 10^7 b^
Mean		5.1 × 10^7^	1.2 × 10^5^	2.2 × 10^8^	2.7 × 10^7^
Commercial	Winter	2.5 × 10^7 a^	4.3 × 10^3 b^	2.0 × 10^7 b^	1.4 × 10^8 a^
	Spring	5.7 × 10^8 b^	5.9 × 10^2 b^	1.6 × 10^8 a^	2.1 × 10^6 a^
	Summer	9.2 × 10^10 c^	1.9 × 10^3 a^	1.3 × 10^8 a^	3.0 × 10^6 b^
	Autumn	1.9 × 10^8 a^	7.2 × 10^3 c^	2.7 × 10^8 b^	5.0 × 10^4 c^
Mean		2.3 × 10^10^	3.5 × 10^3^	1.5 × 10^8^	3.6 × 10^7^

^a,b,c^—different in the column for each of the two analyzed breeding systems; significance was set at the value of 0.05.

**Table 3 animals-12-00740-t003:** Average microbial contamination (CFU/g) of settled dust collected from outside the poultry house only for commercial farms; for backyard farms, no results were recorded.

Type of Poultry Production	Season	TBC	*Enterobacteriaceae*	*Pseudomonas*	The Number of Molds and Yeasts
Commercial	Winter	6.3 × 10^8 b^	1.1 × 10^2 a^	1.4 × 10^6 a^	1.1 × 10^4 a^
	Spring	1.0 × 10^8 a^	1.3 × 10^2 a^	7.4 × 10^7 b^	1.3 × 10^5 b^
	Summer	5.1 × 10^8 b^	1.9 × 10^3 a^	9.5 × 10^7 b^	3.9 × 10^4 a^
	Autumn	1.7 × 10^8 a^	6.5 × 10^3 b^	1.6 × 10^8 c^	6.1 × 10^7 c^
Mean		3.5 × 10^8^	2.1 × 10^3^	8.3 × 10^7^	1.5 × 10^7^

^a,b,c^—different in the column for each of the two analyzed breeding systems; significance was set at the value of 0.05.

**Table 4 animals-12-00740-t004:** Average microbial contamination (CFU/g) of the liter.

Type of Poultry Production	Season	TBC	*Enterobacteriaceae*	*Pseudomonas*	The Number of Molds and Yeasts
Backyard	Winter	8.0 × 10^9 b^	1.1 × 10^5 a^	1.8 × 10^10 b^	2.8 × 10^5 a^
	Spring	8.2 × 10^10 b^	3.5 × 10^5 b^	4.8 × 10^6 a^	1.2 × 10^6 b^
	Summer	6.3 × 10^10 a^	3.3 × 10^4 b^	1.8 × 10^6 a^	3.2 × 10^6 b^
	Autumn	5.8 × 10^9 a^	1.8 × 10^6 a^	1.4 × 10^8 a^	2.6 × 10^8 c^
Mean		4.0 × 10^10^	5.7 × 10^5^	4.5 × 10^9^	6.6 × 10^7^
Commercial	Winter	8.9 × 10^10 c^	5.3 × 10^5 c^	3.2 × 10^10 c^	4.6 × 10^6 c^
	Spring	1.6 × 10^10 a^	1.3 × 10^4 a^	8.9 × 10^9 b^	4.0 × 10^4 a^
	Summer	3.3 × 10^11 b^	3.6 × 10^3 b^	8.2 × 10^9 b^	1.5 × 10^4 a^
	Autumn	2.0 × 10^10 a^	5.3 × 10^4 c^	3.8 × 10^7 a^	2.3 × 10^5 b^
Mean		1.1 × 10^11^	1.5 × 10^5^	1.2 × 10^10^	1.2 × 10^6^

^a,b,c^—different in the column for each of the two analyzed breeding systems; significance was set at the value of 0.05.

**Table 5 animals-12-00740-t005:** Average microbial contamination (CFU/g) of the feed.

Type of Poultry Production	Season	TBC	*Enterobacteriaceae*	*Pseudomonas*	The Number of Molds and Yeasts
Backyard	Winter	4.1 × 10^10^	2.7 × 10^4^	9.0 × 10^7^	2.3 × 10^5^
	Spring	5.4 × 10^11^	2.8 × 10^2^	1.6 × 10^10^	1.2 × 10^5^
	Summer	7.1 × 10^7^	2.9 × 10^4^	5.0 × 10^7^	1.4 × 10^4^
	Autumn	5.5 × 10^6^	2.3 × 10^3^	2.9 × 10^8^	2.7 × 10^6^
Mean		1.5 × 10^11^	1.5 × 10^4^	4.1 × 10^9^	7.7 × 10^5^
Commercial	Winter	4.5 × 10^10^	2.7 × 10^4^	8.0 × 10^4^	4.6 × 10^3^
	Spring	2.3 × 10^7^	3.5 × 10^2^	4.7 × 10^6^	7.4 × 10^6^
	Summer	1.7 × 10^7^	5.3 × 10^1^	8.8 × 10^9^	6.0 × 10^3^
	Autumn	4.5 × 10^6^	7.3 × 10^1^	3.9 × 10^5^	3.3 × 10^3^
Mean		1.1 × 10^10^	6.9 × 10^3^	2.2 × 10^9^	1.9 × 10^6^

**Table 6 animals-12-00740-t006:** Differences for the mean values of individual groups of compounds depending on the breeding system.

	Backyard	Commercial
	Concentration (μg/m^3^)	Concentration (μg/m^3^)
Alcohols	76.58 ^a^	87.93 ^b^
Aldehydes	39.09 ^a^	46.12 ^b^
Ketones	136.35 ^a^	174.39 ^b^
Hydrocarbons	42.92 ^a^	35.31 ^a^
Acids	153.00 ^a^	172.35 ^b^
Terpenes	23.60 ^a^	29.49 ^a^
Benzene and its derivatives	20.52 ^a^	17.14 ^a^
Phenols	41.95 ^a^	48.16 ^a^
Sulfur-containing compounds	39.79 ^a^	60.19 ^b^
Othere	40.24 ^a^	56.12 ^b^

^a,b^—different in the row for each of the two analyzed breeding systems; significance was set at the value of 0.05.

**Table 7 animals-12-00740-t007:** Summary for averages of particular groups of chemical compounds in particular seasons of the year, depending on the type of rearing, taking into account the standard deviation.

Range Mean Concentration ± SD (μg/m^3^)	Backyard				Commercial			
	Winter	Spring	Summer	Autumn	Winter	Spring	Summer	Autumn
Alcohols	58.1 ± 72.4 ^a^	65.3 ± 80.1 ^b^	89.6 ± 106.7 ^c^	93.4 ± 110.2 ^d^	64.4 ± 70.7 ^a^	75.7 ± 78.7 ^b^	106.4 ± 104.9 ^c^	105.2 ± 105.3 ^c^
Aldehydes	29.9 ± 27.6 ^a^	34.7 ± 31.1 ^a^	44.8 ± 36.7 ^b^	47.0 ± 37.9 ^b^	33.0 ± 24.5 ^a^	40.7 ± 28.9 ^a^	59.1 ± 46.7 ^b^	51.7 ± 31.6 ^b^
Ketones	99.1 ± 244.7 ^a^	114.6 ± 198.4 ^b^	163.1 ± 262.6 ^c^	168.6 ± 268.6 ^c^	123.1 ± 185.2 ^a^	147.9 ± 224.8 ^b^	214.8 ± 293.7 ^c^	211.8 ± 294.0 ^c^
Hydrocarbons	30.4 ± 44.9 ^a^	36.2 ± 51.0 ^a^	51.4 ± 65.6 ^b^	53.6 ±67.0 ^b^	22.9 ± 18.5 ^a^	29.7 ± 20.6 ^a^	45.2 ± 29.4 ^b^	43.4 ± 30.8 ^b^
Acids	113.2 ± 244.7 ^a^	129.3 ± 279.5 ^b^	181.8 ± 358.1 ^c^	187.8 ± 366.4 ^c^	126.1 ± 245.8 ^a^	145.1 ± 281.7 ^b^	212.1 ± 369.3 ^c^	206.1 ± 370.0 ^c^
Terpenes	13.1 ± 9.3 ^a^	16.4 ± 11.0 ^a^	30.6 ± 25.0 ^b^	34.3 ± 28.6 ^b^	15.6 ± 4.6 ^a^	21.8 ± 7.2 ^a^	41.3 ± 23.7 ^b^	39.3 ± 23.1 ^b^
Benzene and its derivatives	13.8 ± 10.2 ^a^	17.6 ± 14.3 ^a^	24.4 ± 19.0 ^b^	26.3 ± 19.4 ^b^	12.0 ± 9.5 ^a^	14.9 ± 11.2 ^a^	20.4 ± 14.5 ^a^	21.2 ± 14.2 ^a^
Phenols	27.2 ± 23.0 ^a^	33.8 ± 23.3 ^a^	50.5 ± 33.8 ^b^	56.4 ± 44.7 ^b^	35.2 ± 15.6 ^a^	38.5 ± 21.1 ^a^	60.6 ± 28.3 ^b^	58.3 ± 29.4 ^b^
Sulfur-containing compounds	28.2 ± 26.3 ^a^	32.4 ± 30.0 ^a^	48.4 ± 41.5 ^b^	50.2 ± 42.4 ^b^	39.1 ± 26.9 ^a^	50.6 ± 25.8 ^b^	83.3 ± 63.9 ^c^	67.7 ± 44.3 ^b^
Othere	29.3 ± 30.6 ^a^	33.4 ± 35.0 ^a^	48.4 ± 46.9 ^b^	49.9 ± 47.8 ^b^	39.2 ± 25.7 ^a^	47.4 ± 32.3 ^a^	70.2 ± 47.3 ^b^	67.8 ± 45.7 ^b^

^a,b,c,d^—different in the row for each of the two analyzed breeding systems; significance was set at the value of 0.05.

## Data Availability

Anonymised data are available from the authors on reasonable request.

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
