# Peer review of "Comprehensive Assessment of Environmental Pollution in a Poultry Farm Depending on the Season and the Laying Hen Breeding System"

_animals, 2022, doi:10.3390/ani12060740_

Round 1
Reviewer 1 Report
The paper entitled “Comprehensive assessment of environmental pollution in a poultry farm depending on the season and the laying hen breeding system” by Szablewskiet al. addresses an interesting and important subject. The title of the article is correct and is consistent with its content. Moreover, the laboratory methodology is correct. However, the manuscript requires a major revision before it can be recommended for publication.
The specific comments are included in the attached file.

Author Response
Reviewer Comments
The paper entitled “Comprehensive assessment of environmental pollution in a poultry farm
depending on the season and the laying hen breeding system” by Szablewskiet al. addresses an
interesting and important subject. The title of the article is correct and is consistent with its
content. Moreover, the laboratory methodology is correct. However, the manuscript requires a
major revision before it can be recommended for publication.
Answer: Thank you very much for valuable comments, thanks to them the article gained scientific value. This manuscript takes into account all comments and suggestions from the reviewer.
Specific comments:
Simple summary:
According to the instructions for author of Animals journal:
Simple Summary: It is vitally important that scientists are able to describe their work simply and concisely to the public, especially in an open-access on-line journal. The simple summary consists of no more than 200 words in one paragraph and contains a clear statement of the problem addressed, the aims and objectives, pertinent results, conclusions from the study and how they will be valuable to society. This should be written for a lay audience, i.e., no technical terms without explanations. No references are cited and no abbreviations. Submissions without a simple summary will be returned directly. Example could be found at https://www.mdpi.com/2076-2615/6/6/40/htm.
Answer: The abstract has been corrected
Abstract:
According to author instructions of Animals journal, the abstract is about 200 words maximum.
Please, adequate its length.
Answer: The abstract has been corrected
Introduction:
Lines 71-81: this paragraph should be moved to the materials and methods section. It could be an
introduction paragraph before the details of the project.
Answer: Moved according to reviewer's suggestion
Moreover, the change of the experimental groups denomination throughout all the manuscript
is very confusing. Please, unify.
Answer: Unify according to reviewer's suggestion
Materials and methods:
This section should be re-written. The subsubsections should be re-organized.
I consider that it should be more or less like this scheme:
- Materials and methods
- Paragraph of the introduction adapted to the new location.
2.1 Test material or experimental design
2.2 Sampling methods (the original title of this subsubsection was wrong)
2.2.1 Feed and bedding intake
2.2.2 Dust collection
2.2.3 Volatile metabolites
- In these subsubsections only must be explained the materials and methods about sample
collection.
2.3 Laboratorial analysis
- Add an introduction paragraph about the samples analysis, maybe with the information of the
previous paragraphs, but adapted to this new location.
2.3.1 Total number of bacteria
2.3.2 Enterobacteriaceae count
2.3.3 The number of molds and yeast
2.4 Survey methods
2.5 Statistical analysis
Answer: Re-organized and corrected according to reviewer's suggestion
Line 96: what is the meaning of n=16 and n=8?
Answer: This is the number of poultry houses tested corrected according to reviewer's suggestion
Line 102: only the farms from the first experimental group are represented at Table 1? Table 1
should be better indicated on the text.
Answer: Corrected according to reviewer's suggestion
Line 105: 4 large-scale farms were sampled? This data is different from data showed on Table 1.
Answer: Corrected according to reviewer's suggestion
Lines 107-109...: please, review the superscripts throughout all the manuscript (including tables).
Answer: Corrected according to reviewer's suggestion
Table 1: if the experimental groups are named commercial and backyard farms, unify with the
text.
Answer: Unify according to reviewer's suggestion
Lines 115-120: results obtained for the surveys (340 residents, 150 people, and women and men
percentages) should be indicated in the results section, not in materials and methods section.
Answer: Moved according to reviewer's suggestion
Lines 126 (since “Individual”)-128: this information should be moved to laboratorial analysis.
Answer: Moved according to reviewer's suggestion
Lines 136 (since “Volatile...”)-141: this information should be moved to laboratorial analysis.
Answer: Moved according to reviewer's suggestion
Line 143: “samples weighing 20 g were taken” should be moved to sampling methods.
Answer: Moved according to reviewer's suggestion
Lines 146-147: what is the “dilution fluid”? What is the commercial reference?
Answer: It was mistake, it shoud be: “diluting fluid”, corrected and completed according to reviewer's suggestion
Line 150: please, add the commercial reference of the agar medium.
Answer: Moved and completed according to reviewer's suggestion
Line 160: if the method and the medium was the same for all the bacteria and for enterobacteria,
what is the difference? Please, reference the analysis methods.
Answer: Completed according to reviewer's suggestion
Line 162: “Pseudomonas” must be written in italic. This mistake is repeated in different parts of
the manuscript, please verify.
Answer: Verified and completed according to reviewer's suggestion
Line 163-168: please, describe the analysis methods.
Answer: Completed according to reviewer's suggestion
Line 172: the number of respondents should be moved to the results section.
Answer: Moved and completed according to reviewer's suggestion
Results:
Line 190: this subsection is incorrect, in the results section you cannot discuss the results, the
discussion has its own section in the manuscript.
Answer: Completed according to reviewer's suggestion
Line 190: before the different results obtained, an introduction paragraph including the total
number of farms sampled and surveys performed should be added.
Answer: Completed according to reviewer's suggestion
Line 191: the subsections of the results section should be exposed in the same order of the
objectives and the materials and methods sections.
Answer: Corrected according to reviewer's suggestion
Lines 199-203: “On the basis of the identified compounds, they were assigned to 10 groups: alcohols,
aldehydes, ketones, hydrocarbons, acids, terpenes, benzene and its derivatives, phenols, and sulfur-
containing compounds. The remaining volatile compounds that do not belong to any of the above-
mentioned groups were combined into the "other" group” this information should be moved to
materials and methods section.
Answer: Corrected and moved according to reviewer's suggestion
Lines 220-222: this information should be moved to discussion section.
Answer: Moved according to reviewer's suggestion
Table 2: decimal numbers must be separated by dots.
Answer: Corrected according to reviewer's suggestion
Table 2: please, correct superscripts.
Answer: Corrected according to reviewer's suggestion
Table 3: please, center Backyard and Commercial tittles, and separate better the data presented,
it is a little bit confusing.
Answer: Corrected according to reviewer's suggestion
Table 3: decimal numbers must be separated by dots.
Answer: Corrected according to reviewer's suggestion
Table 3: please, correct superscripts.
Answer: Corrected according to reviewer's suggestion
Lines 245-248: “Their presence in the atmosphere of the henhouse has not been studied so far. Among them, trichodiene plays an important role, due to the fact that it is considered a precursor of the formation of mycotoxins from the group of trichothecenes, mainly deoxynivalenol” should be moved to the discussion section.
Answer: Moved according to reviewer's suggestion
Figure 2: please, specify the group of each color (black and white).
Answer: Added according to reviewer's suggestion
Lines 281-288: please, correct the superscripts.
Answer: Corrected according to reviewer's suggestion
Table 4: there are not statistical differences?
Answer: Added according to reviewer's suggestion
Lines 292-296: this information should be moved to materials and methods section.
Answer: Moved according to reviewer's suggestion
Table 5: if there is not information about backyard group, please, specify it on the title of the table
and delete backyard information.
Answer: Corrected according to reviewer's suggestion
Line 315 and 318: “Pseudomonas” must be written in italic.
Answer: Corrected according to reviewer's suggestion
Line 318: superscripts.
Answer: Corrected according to reviewer's suggestion
Table 6: there are not statistical differences?
Answer: Added according to reviewer's suggestion
Line 328: what is the meaning of OLB?
Answer: It was mistake, it shoud be: “TBC”, corrected and completed according to reviewer's suggestion
Line 330: “Enterobacteriaceae” must be written in italic.
Answer: Corrected according to reviewer's suggestion
Table 7: there are not statistical differences?
Answer: Added according to reviewer's suggestion
Lines 341-366: what are the results obtained? Percentages?
Answer: Yes in percentages, corrected according to reviewer's suggestion
Discussion:
Line 419: delete (.
Answer: Corrected according to reviewer's suggestion
Line 426: superscript.
Answer: Corrected according to reviewer's suggestion
Line 432: superscript.
Answer: Corrected according to reviewer's suggestion
Line 441-460: the results must be compared with those obtained in other studies.
Answer: Added and corrected according to reviewer's suggestion
Conclusion:
Lines 465: please, specify the information about the impact of the season on the microbiological
contamination.
Answer: Added and corrected according to reviewer's suggestion
Lines 481-487: this information should be moved to the discussion section.
Answer: Moved according to reviewer's suggestion
Lines 495-496: this sentence is from the template; it must be removed for the manuscript.
Answer: Removed according to reviewer's suggestion
References:
Please, check the references formats to adequate it to Animals journal.
Answer: Formatted according to reviewer's suggestion
Reviewer 2 Report
The article takes into consideration intersting issues and concerns about intensive and extensive poultry farms. The introduction is clear and also objectives are pretty clear. Materials and method are well described. The problems begin with results. In the first part part, from line 192 to line 212 everithing is clear.
From line 212 to 216 the authors in the description of differences omit to say which one of the two system is better or worse.
From line 236 to line 243: all clear
From line 244 to line 248: It's not clear why among all the compounds the authors chose to describe in detail only trichodiene. Trichodien is condidered a precursor or it is "universally known" as a precursor?The first statement needs references, the second don't.
From line 248 to line 251: Significative differences beetween the two systems are only in Autumn. The total amount is comparable with other studies on health safety? Without other data it's impossible to say if it can be a danger for animals and humans. Trichodien is the common volatile precursor during the biosynthesis of trichothecenes, trichodiene is considered to be a biomarker for the respective mycotoxin content in feed samples. Did the author considered to compare their results with other studies about feed contamination? The described amounts are an hazard to feed safety?
Figure 2: legenda is missing
Figure 3: the A and B chromatograms are not comparable, the second seems to be a zoom, time release is missing and there aren't informations about the scale. Why is the arrow in A is pointed on 21.96? A abd B are not comparable if the figures are different.
Microbiological contamination: the statistic cited in materials and method did not appear in this section. The results are only described but without statistic significance they are not comparable.
Survey results in general seems a litlle confused. It is not clear what answer were made and what intended for "pollution" or if the authors explained the goal of the survey. In addition the missing of tables or figures make the study less interesting.
Discussion: the first part of the discussion is not based on the results and most references are in Polish.
Until line 424 the discussion is too general and seems more like an introduction.
From line 425 to line 431: significance is mentioned abot microbiological contaminations. Those data are not showed in results.
Conclusions: beeing the discussion not decisive to establish which one of the two system was better, the conclusions are not aligned with the discussion.
References: bibliografy is scarce. Out of 23 studies, 9 are in Polish, 3 are about methods and 5 seem to be not referenced. The subject matter is of paramount importance and needs to be compared with a richer bibliografy.
Author Response
Answer to the reviewer:
Thank you very much for your valuable tips and comments. They have all been taken into account and revised meticulously. Below are the responses to the reviewer's comments.
The article takes into consideration intersting issues and concerns about intensive and extensive poultry farms. The introduction is clear and also objectives are pretty clear. Materials and method are well described. The problems begin with results. In the first part part, from line 192 to line 212 everithing is clear.
From line 212 to 216 the authors in the description of differences omit to say which one of the two system is better or worse.
Answer: Completed: "On the basis of the obtained quantitative results of volatile compounds in the air, it can be concluded that commercial farming is worse. However, the qualitative picture of volatile compounds indicates a greater diversity in their occurrence in backyard farming, which carries a greater risk."
From line 236 to line 243: all clear
Answer:ok.
From line 244 to line 248: It's not clear why among all the compounds the authors chose to describe in detail only trichodiene. Trichodien is condidered a precursor or it is "universally known" as a precursor?The first statement needs references, the second don't.
Answer: Literature was supplemented. Since 1995, Trichodiene is considered a precursor in the biosynthesis of mycotoxins from the group of trichothecenes, but is also considered a biomarker of the presence of mycotoxins in the studied environment.
From line 248 to line 251: Significative differences beetween the two systems are only in Autumn. The total amount is comparable with other studies on health safety? Without other data it's impossible to say if it can be a danger for animals and humans. Trichodien is the common volatile precursor during the biosynthesis of trichothecenes, trichodiene is considered to be a biomarker for the respective mycotoxin content in feed samples. Did the author considered to compare their results with other studies about feed contamination? The described amounts are an hazard to feed safety?
Answer:The trichodiene itself is not a threat, but its presence does indicate the possibility of the presence of mycotoxins from the trichothecening group. No mycotoxins were identified in the feed samples tested and the results were not included in the publication. There was no publication on trichodiene and its compound content in feed. Reports on the amount of trichodiene in cereal grains. Supplemented literature.
Figure 2: legenda is missing
Answer: legend added.
Figure 3: the A and B chromatograms are not comparable, the second seems to be a zoom, time release is missing and there aren't informations about the scale. Why is the arrow in A is pointed on 21.96? A abd B are not comparable if the figures are different.
Answer: The figure was corrected and completed.
Microbiological contamination: the statistic cited in materials and method did not appear in this section. The results are only described but without statistic significance they are not comparable.
Answer: The statistics were supplemented in the microbiological results.
Survey results in general seems a litlle confused. It is not clear what answer were made and what intended for "pollution" or if the authors explained the goal of the survey. In addition the missing of tables or figures make the study less interesting.
Answer: Corrected, supplemented. and added some charts (Figore 3).
Discussion: the first part of the discussion is not based on the results and most references are in Polish.
Answer: English literature added.
Until line 424 the discussion is too general and seems more like an introduction.
Answer: Corrected
From line 425 to line 431: significance is mentioned abot microbiological contaminations. Those data are not showed in results.
Answer: Corrected
Conclusions: beeing the discussion not decisive to establish which one of the two system was better, the conclusions are not aligned with the discussion.
Answer: Corrected
References: bibliografy is scarce. Out of 23 studies, 9 are in Polish, 3 are about methods and 5 seem to be not referenced. The subject matter is of paramount importance and needs to be compared with a richer bibliografy.
Answer: Corrected, completed.
Round 2
Reviewer 1 Report
Upon review, I recommend this manuscript for publication in Animals Journal.
Author Response
Thank you very much for your positive review
Reviewer 2 Report
My previous questions and new comments
From line 212 to 216 the authors in the description of differences omit to say which one of the two system is better or worse.
Answer: Completed: "On the basis of the obtained quantitative results of volatile compounds in the air, it can be concluded that commercial farming is worse. However, the qualitative picture of volatile compounds indicates a greater diversity in their occurrence in backyard farming, which carries a greater risk."
Maybe those comment should be included in the text otherwise the reader could think that you have a bias toward small farms
Line 258: legend for White and Black columns is missing (already requested)
Lines 261 and 270: Figure 3: A and B chromatograms are not comparable, the second seems to be a zoom, time release is missing and there aren't information about the scale. Why is the arrow in A pointed on 21.96? A and B are not comparable if the figures are different.
Answer: The figure was corrected and completed. (I don’t see these corrections)
Microbiological contamination: the statistic cited in materials and method did not appear in this section. The results are only described but without statistic significance they are not comparable.
Answer: The statistics were supplemented in the microbiological results. (I don’t see these corrections in the tables and in addition decimal sign must be a dot.)
From line 425 to line 431: significance is mentioned abot microbiological contaminations. Those data are not showed in results.
Answer: Corrected . (I don’t see these corrections)
